# The long shadow of 9/11: Mental health outcomes in adult children of World Trade Center Responders with PTSD

Yael M. Cycowicz[1,2]*, Diana V. Rodriguez-Moreno[1,2], Daniel Craft[1], Keely Cheslack-Postava[1,2]

1 New York State Psychiatric Institute, New York, New York, United States of America, 2 Department of Psychiatry, Columbia University, New York, New York, United States of America

* yc60@cumc.columbia.edu

## Abstract

Intergenerational transmission of trauma (ITT) has been documented among offspring of parents exposed to war, genocide, and interpersonal violence, yet little is known about ITT among families of World Trade Center responders (WTC-R). The aftermath of 9/11 terrorist attacks provides a unique naturalistic context for examining long-term psychological outcomes among now-adult offspring raised during periods of parental trauma exposure and recovery. The WTC-R cohort comprised traditional first responders (i.e., law enforcement officers) and nontraditional recovery workers, among whom those with PTSD were invited to complete an online survey and recruit their adult offspring. Parents and offspring completed standardized assessments of mental health, including depression, anxiety, PTSD, panic disorder, substance use, and lifetime traumatic experiences, as well as measures of social support, family relationships, and quality of life. Generalized Estimating Equations (GEE) models were used to examine associations between parental 9/11-related exposure, current parental psychopathology, and offspring mental health outcomes, with moderation by responder occupational category. Greater parental 9/11 exposure was associated with a higher likelihood of PTSD, anxiety, and panic symptoms among adult offspring. Current parental PTSD and depression were significantly associated with offspring PTSD and depression, respectively. Poor parent-offspring relationships were associated with offspring depression, PTSD, and alcohol use disorder. Several associations were moderated by occupational subgroup, suggesting differential vulnerability linked to the nature and context of exposure. These findings demonstrate enduring intergenerational mental health effects more than two decades after 9/11, highlighting how parental trauma exposure, ongoing psychopathology, and family relational functioning collectively shape outcomes in WTC-R offspring.

**Data availability statement:** Our study includes DUA, which states legal and privacy restrictions. Therefore, data will be made available under legally required restrictions. A minimum data set can be requested via email to itt-data-request@nyspi.columbia.edu. Institute staff, including IT, research administration, and/or study personnel, will review the request and initiate data use/sharing processes based on the request details and data access terms, including those from study sponsors and partners, including the WTC Health Program GRDC.

**Funding:** This work was supported by the National Institute for Occupational Safety and Health (NIOSH)/ Centers for Disease Control and Prevention (CDC), under Grant No. U01OH012065 to YMC, DVRM, and KCP. The funders had no role in study design, data collection and analysis, decision to publish, or preparation of the manuscript. All authors received a salary from this grant.

**Competing interests:** The authors have declared that no competing interests exist.

## Introduction

Exposure to violence, particularly in the context of man-made mass trauma, is a well-established risk factor for the development of psychiatric disorders in affected individuals [1]. However, the repercussions of such trauma often extend beyond the directly exposed individual, influencing the next generation through a phenomenon known as the Intergenerational Transmission of Trauma (ITT) [2,3]. While ITT has been widely documented in the descendants of Holocaust survivors [4,5] and war veterans [3,6], relatively little is known about ITT among the children of first responders (FR) [7], especially those involved in the aftermath of the terrorist attacks on the World Trade Center (WTC) on September 11, 2001 [8].

On 9/11 and up to a year following the attack, tens of thousands of traditional and nontraditional responders were deployed to Ground Zero, collectively referred to as the World Trade Center Responders (WTC-R). They included law enforcement officers, firefighters, emergency medical technicians (EMTs), and a large cohort of civilian recovery and cleanup workers. WTC-R were repeatedly exposed to gruesome scenes of destruction and human loss, placing them at high risk for post-traumatic stress disorder (PTSD), depression, anxiety, and substance use disorders (SUD) [9,10], as well as for physical health issues due to exposure to toxicants [11]. Thus, the potential for 9/11 trauma to be transmitted across generations presents serious implications for public health and mental health policy.

Several factors of trauma-exposed parents have been identified as modifiers or mediators of ITT, including exposure duration [12], gender [13], ethnicity [14], parental psychiatric status [15], quality of family relationships, and availability of social support [16]. Parents with PTSD or other psychiatric disorders may struggle with emotional availability, inconsistent caregiving, a deficit in managing conflicts, or creating environments that undermine children's emotional and cognitive development. Additionally, trauma-exposed parents often experience difficulties in communicating their emotions or trauma histories [17], which may further isolate children and impede the development of trust and emotional security. Thus, trauma-exposed parents may transmit psychological distress to their offspring not only through genetic vulnerability but also through disrupted parenting [18], altered family dynamics, and modeling of maladaptive coping mechanisms [19].

Most of the research on parental trauma exposure has focused narrowly on PTSD. However, trauma often results in broader cognitive, emotional, and psychosocial impairments, and the scope of ITT is likewise more expansive. For children, these transgenerational effects may manifest as emotional dysregulation [20,21], behavioral challenges, academic difficulties, and increased risk for developing psychiatric disorders, including anxiety, depression, PTSD, and SUD [19,22]. Garagano et al. [23], for example, reported that 9/11 exposed parents who reported a greater number of days with poor mental health were more likely to have adolescents exhibiting behavioral and emotional difficulties.

Existing ITT research has primarily examined children indirectly exposed to trauma through the emotional and behavioral consequences of their parents' traumatic experiences, especially where those traumatic events occurred in the parents' early

life or in combat settings before the children were born [4,24]. The WTC-R cohort presents a unique opportunity to investigate ITT among children who were born before the 9/11 terror attacks, who therefore experienced the effects of parental trauma during the time of exposure, and likely grew up in households changed and shaped by *ongoing* trauma and its sequelae, including the development of psychiatric disorders and chronic health problems. Moreover, the WTC-R cohort is composed of two distinct subgroups: traditional FR, such as law enforcement officers (WTC-pR) and firefighters, and non-traditional civilian recovery workers (WTC-wR) who labored for extended periods at Ground Zero with minimal training or institutional support. Despite experiencing similar trauma exposures, these two groups differ significantly in occupational roles, socioeconomic status, and access to social support, factors that are critical to understanding differential vulnerability within the WTC-R population and, by extension, their families. For instance, in the early years after 9/11, PTSD prevalence among WTC-wR was four times higher than among WTC-pR [25], suggesting that occupational role and associated psychosocial factors may influence resilience and recovery. Finally, the mental health burden on WTC-R has not only been persistent but has also shown concerning trajectories over time [26]. Whereas PTSD symptoms in WTC-pR tended to stabilize or decrease in the years following the event, possibly due to peer support and occupational culture, WTC-wR experienced worsening trajectories of psychological symptoms [10]. These distinct long-term consequences suggest differential effects on ITT between the two WTC groups and raise important questions about their influence on WTC-R parenting capacity, family dynamics, and ultimately, the psychological development of their children.

Mental health vulnerability to ITT also depends on child-related factors. For example, the children's age at the time of parental trauma, as well as their lifetime exposure to stressors, likely shapes the degree and nature of ITT effects. Studies have shown that younger children may exhibit more externalizing symptoms such as aggression and impulsivity, [27] while adolescents may internalize distress in the form of depression and withdrawal [28]. Furthermore, children with a history of prior traumatic exposures are particularly vulnerable to reactivation or exacerbation of psychiatric symptoms when confronted with new stressors, as seen in the increased PTSD prevalence among individuals who were exposed to both 9/11 and Hurricane Sandy [29].

In light of these findings, the current study aims to comprehensively assess ITT in a new cohort of now-adult children of WTC-R, with a focus on comparing outcomes between offspring of WTC-pR and WTC-wR. The study utilizes self-report measures of WTC-R parents and at least one of their now-adult children. WTC-R parents were recruited via the General Responder Data Center (GRDC), which maintains longitudinal health data on WTC-R from the WTC Health Program (WTCHP) [30]. Only WTC-R who were documented as parents on 9/11/2001 and had a diagnosis of PTSD in at least one prior WTCHP clinical visit were eligible for recruitment. By leveraging this uniquely well-documented cohort, we aim to identify the emotional and psychiatric correlates of ITT, as well as factors related to transmission.

## Methods

### Ethical statement

The study was approved by the NYSPI Institutional Review Board (protocol #8009), and all participants were required to review the consent form and to indicate their agreement to participate before proceeding to complete the online survey.

### Participants and recruitment

WTC-R parents were recruited through the World Trade Center Health Program (WTCHP) General Responders Data Center (GRDC) at the Icahn School of Medicine at Mount Sinai. The WTCHP, through its Clinical Centers of Excellence and GRDC, manages physical and mental health, exposure, occupational, and socioeconomic data of non-FDNY responders. Our cohort includes both traditional FR (e.g., law enforcement officers; WTC-pR) and non-traditional FR (e.g., utility, sanitation, and construction workers; WTC-wR) involved in 9/11 rescue and recovery. Eligibility criteria were a) PTSD diagnosis by the Diagnostic Interview Schedule at one or more WTCHP visits, b) reporting at their first WTCHP

screening that they had children under the age of 18 on 9/11/2001, c) consent to be contacted by external researchers, and d) availability of data from at least 4 WTCHP visits, with the most recent visit occurring after 2015. The GRDC provided contact information for all eligible WTC-R. Each received an invitation email describing the online self-report survey, noting that participation from at least one of their children is required, and indicating $30 compensation per participant. Individuals without an email address were contacted by phone to obtain one. Each invitation included a unique personalized link.

Now-adult children were recruited through their participating WTC-R parent. After a parent enrolled (Fig 1), they entered each child's name, age, and contact information. Each child then automatically received an email with study information and a unique link to enroll and complete the survey. Once at least one child completed the survey, the parents received a follow-up email with a new link to complete their own final survey. This process ensured that we obtained data from at least one child for each participating parent.

## Data collection and procedures

The remote online study included a REDCap-based survey and a short battery of cognitive tasks administered online via Gorilla®. Data were collected from April 20, 2021, to October 6, 2022. This paper reports only data from the REDCap instruments, which are not diagnostic and offer an indication of probable mental health disorders.

## Instruments

Aside from the initial registration step in which WTC-R parents provided contact information of their now-adult children, parents and children completed largely the same survey (with minor differences noted below).

The survey assessed demographics, mental health, physical health, and family and social support. Table 1 presents the list of instruments used. Information on parental exposure to the 9/11 attacks and WTC-R roles was obtained from the GRDC. However, adult children completed a 9/11 exposure module, which was not included in the present analyses.

The PARQ differed slightly for parents and children to reflect their respective relationships. Parents with multiple participating children indicated which child their responses referred to. The 3 items from the MOS are presented in S1 Text. Because data collection occurred during the COVID-19 pandemic, participants completed a pandemic-related experiences module described in the S2 Text, and four PCL items were included to screen for possible pandemic-related PTSD symptoms. Physical health assessment differed by group: parents reported whether each diagnosis occurred before or after 9/11, whereas adult children reported lifetime diagnosis only.

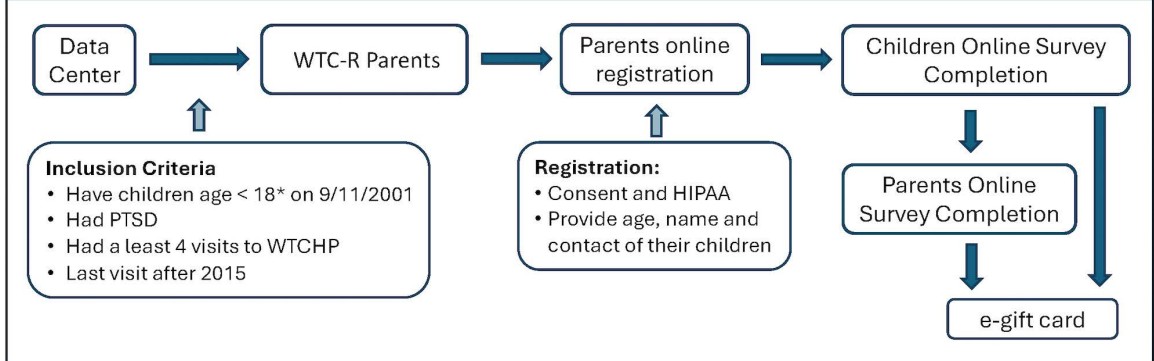

**Fig 1. Study design diagram.** * Data set includes 2 children who were between the ages of 19 and 20 on 9/11/2001.

**Table 1. List of assessment instruments.**

| Domain | Assessment of | Instrument Name |
|---|---|---|
| *Demographic* | Age, sex, race, ethnicity, education, marital status, No. of children, employment | |
| *Mental Health* | PTSD | PCL-5 [31]; Covid-PCL [31] |
| | Depression | PHQ-8 [32] |
| | Anxiety | GAD-7 [33] |
| | Panic | PDSR [34] |
| | Substance Use (AUD, SUD) | Audit [35], DAST-10 [36] |
| *Traumatic Life Event* | No. of life stressors | LEC-5 [31] |
| *Physical health* | List of physical condition | WTC registry [37] |
| *Family and social support* | General social Support Perceived support from spouse, family and friends | 3 items from MOS [38] Perceived Social Support [39] |
| *Parenting Style* | Parent/ child relationship | PARQ [40] |
| *Quality of life* | Life satisfaction | Q-LES-SF [41] |
| *Resilience* | Coping with stress | CD-RISC-2 [42] |
| *WTC Exposure & experience* | Adult-Children only | Adapted from Hoven at al. [43] |
| *Covid-19* | Covid diagnosis in self and family and friends | Developed by our research team specifically for this study |

## Exposure to 9/11

For the WTC-R participants, occupational and 9/11 exposure data were obtained from the GRDC, including both the type and duration of exposure to the 9/11 attack. These variables capture the intensity and nature of responders' exposure, which may relate to long-term health outcomes in their children. We utilized the following GRDC variables (or composites derived from them): a) Arrival time at WTC site; b) Exposure to the dust cloud, due to its known health risks; c) Ever having worked on the pile or in the pit; d) Exposure to human remains; e) Proximity to Ground Zero on 9/11 (indicated by the "South of Canal Street" variable); f) Exposure was rated on a 0 – 3 scale, based on time at the WTC site, dust cloud exposure, and work on the debris pile. The scale was defined as follows: 3 – worked > 90 days, exposed to the dust cloud, and worked on the pile; 2 – exposed to the dust cloud but worked < 90 days or did not work on the pile; 1 – not exposed to the dust cloud and worked 40 – 90 days or did not work on the pile; 0 – worked < 40 days, not exposed to dust, and did not work on the pile; [44] g) Total duration of exposure (in months); and h) Hours categorized as cumulative working hours spent within the first 2 days, days 3 – 7, or days 8–20 post-attack.

## Statistical analysis

We first calculated descriptive statistics summarizing socio-demographic characteristics, prevalence of mental and physical health disorders, and average perceived social support scores separately for parents and their adult children, stratified by parental occupation group (WTC-pR vs. WTC-wR). For parents, we also tabulated 9/11 exposure characteristics. Group differences between parental WTC-pR and WTC-wR were assessed using chi-square tests for categorical variables and t-tests for continuous variables.

We examined the association of parental occupational group, 9/11-related exposures, current mental health, cancer history, trauma exposure, and social support with adult children's current mental health outcomes. Generalized Estimating Equations (GEE) models were used to account for correlation among siblings. Dichotomous outcomes (now-adult

children's depression, GAD, panic, PTSD, and AUD) were modeled using a logit link, and the COVID-PCL (COVID-related PTSD symptoms) was modeled using a linear link. Each parental factor was tested as an independent variable in a separate model. All models were adjusted for the adult child's sex, age, minoritized racial/ethnic status, and the sex of the 9/11-exposed parent. We tested heterogeneity of associations by parent occupational group across all models, by the addition of parent occupational group X parent exposure characteristic, parent mental health, or parent social support product terms to covariate-adjusted models. For models in which the product term was statistically significant ($p < 0.05$), we generated stratum-specific estimates of association by parent occupational group.

To address potential selection bias due to non-participation, we conducted a sensitivity analysis for the GEE models by weighting observations using the inverse probability of participation. Participation probabilities among the eligible population were estimated using a logistic regression model with gender, race, ethnicity, age at 9/11, occupation group, most recent WTCHP visit year, and PTSD status at the most recent WTCHP visit as independent variables.

## Results

### Sample size

E-mail invitations were sent to 2,783 WTC-R parents. Most did not respond to the email or follow-up phone calls or declined participation. Of the 534 parents who consented, 19 indicated no children in the eligible age range, and 188 did not provide information about their children. In some cases, parents wished to participate without involving their children; in others, adult children declined participation. Of the 327 parents who provided child contact information, 270 adult children, from 176 families, completed the survey. The final sample comprises families with one (N = 106), two (N = 60), three (N = 7), and four (N = 3) participating children, as well as 11 children whose parents did not complete the survey.

### Sample demographics

Table 2 summarizes the demographic characteristics of WTC-R parents, stratified by occupational role. A small but statistically significant age difference was observed between groups, with WTC-wR participants being older on average. As expected, the majority of participants were male, consistent with the sex distribution of occupational roles at the time of the 9/11 attacks. The groups appeared similar in their racial and ethnic composition, but significant differences were noted in educational attainment, with a greater proportion of WTC-wR reporting less than a college education. This disparity in education was reflected in corresponding differences in reported household income. In contrast to the parent sample, the demographic characteristics of their children (right panel of Table 2), show a higher proportion of female participants. No significant demographic differences were observed between children of WTC-wR and those of WTC-pR.

### WTC-R 9/11 exposure

The duration and type of exposure experienced by WTC-R participants are presented in Table 3. The majority of WTC-R arrived at the site of the terrorist attack shortly after the event, and exposure to the dust cloud was comparable between the two groups of responders. However, a significantly larger proportion of WTC-pR were present near the 9/11 attack site (south of Canal Street) on September 11, 2001, and were more likely to have encountered human remains. They also remained in the area for longer periods during the following two days, resulting in greater overall exposure.

### Mental health status

Current mental health status is summarized in Table 4 for both WTC-R parents and their adult children (stratified by responder group). As expected, given their longer life span, parents reported a greater number of challenging life events compared to their adult children. Additionally, likely due to the nature of law enforcement work, WTC-pR parents reported significantly more traumatic exposures than WTC-wR parents. Although all parents were selected based on a history of

**Table 2. Demographic characteristics of the WTC-R parents (n = 176) and adult children (n = 270) separately for WTC-wR and WTC-pR groups.**

| | Parents | | | | | | Adult Children | | | | | |
|---|---|---|---|---|---|---|---|---|---|---|---|---|
| | WTC-wR (N = 93) | | WTC-pR (N = 81) | | | | WTC-wR (N = 134) | | WTC-pR (N = 119) | | | |
| | Mean | SD | Mean | SD | p | missing | Mean | SD | Mean | SD | p | missing |
| Age (years) | 60.74 | 5.72 | 58.81 | 5.61 | **0.026** | | 28.69 | 5.29 | 28.66 | 4.92 | 0.952 | |
| | N | % | N | % | | | N | % | N | % | | |
| Sex | | | | | **0.001** | | | | | | 0.747 | 6 |
| Female | 3 | 3.23 | 21 | 25.93 | | | 83 | 63.85 | 73 | 61.86 | | |
| Male | 90 | 96.77 | 60 | 74.07 | | | 47 | 36.15 | 45 | 38.14 | | |
| Ethnicity Spanish speaking | 17 | 18.48 | 14 | 17.50 | 0.868 | 2 | 37 | 28.68 | 36 | 30.25 | 0.786 | 6 |
| Race and ethnicity | | | | | 0.560 | 11 | | | | | 0.548 | 9 |
| Black, non-Hispanic | 2 | 2.30 | 5 | 6.58 | | | 2 | 1.56 | 5 | 4.24 | | |
| Black, Hispanic | 4 | 4.60 | 2 | 2.63 | | | 2 | 1.56 | 2 | 1.70 | | |
| White, non-Hispanic | 61 | 70.12 | 54 | 71.05 | | | 82 | 64.06 | 76 | 64.41 | | |
| White, Hispanic | 10 | 11.49 | 10 | 13.16 | | | 18 | 14.06 | 21 | 17.80 | | |
| Other groups | 8 | 9.20 | 3 | 3.95 | | | 4 | 3.13 | 2 | 1.70 | | |
| More than one group | 2 | 2.30 | 2 | 2.63 | | | 20 | 15.63 | 12 | 10.17 | | |
| Education | | | | | **0.013** | 2 | | | | | 0.147 | 3 |
| HS or less | 22 | 23.91 | 5 | 6.25 | | | 12 | 9.09 | 8 | 6.78 | | |
| Some college/Assoc. degree | 43 | 46.74 | 42 | 52.50 | | | 39 | 29.55 | 26 | 22.03 | | |
| Bachelor's degree | 19 | 20.65 | 21 | 26.25 | | | 47 | 35.61 | 59 | 50.00 | | |
| Graduate degree | 8 | 8.70 | 12 | 15.00 | | | 34 | 25.76 | 25 | 21.19 | | |
| Employed (full or part time) | 35 | 38.89 | 28 | 34.57 | 0.559 | 3 | 107 | 82.31 | 95 | 81.20 | 0.821 | 6 |
| Income | | | | | **0.028** | 3 | | | | | 0.406 | 7 |
| Up to $99,000 | 40 | 43.96 | 18 | 22.50 | | | 70 | 53.85 | 51 | 43.97 | | |
| $100,000-199,999 | 34 | 37.36 | 38 | 47.50 | | | 32 | 24.62 | 37 | 31.90 | | |
| $200,000 or above | 10 | 10.99 | 14 | 17.50 | | | 9 | 6.92 | 7 | 6.03 | | |
| Prefer not to answer | 7 | 7.69 | 10 | 12.50 | | | 19 | 14.62 | 21 | 18.10 | | |
| Married | 64 | 68.82 | 63 | 78.75 | 0.140 | 1 | 35 | 26.12 | 28 | 23.53 | 0.634 | |
| Lives with a partner* | 58 | 62.37 | 67 | 82.72 | **0.003** | | 50 | 37.31 | 49 | 41.18 | 0.530 | |
| Lives with child(ren)* | 42 | 45.16 | 44 | 54.32 | 0.228 | | 27 | 20.15 | 23 | 19.33 | 0.870 | |
| Lives with parents* | | | | | | | 50 | 37.31 | 47 | 39.50 | 0.722 | |

*This is based on a question regarding household size.

PTSD, the majority of the sample did not meet criteria for current PTSD. Instead, we observed a higher prevalence of depression, anxiety, and panic disorder, with no significant differences between responder groups. Among adult children, over 20% had depression, and over one quarter had an anxiety disorder. Importantly, while PTSD was relatively low among the adult-children, they exhibited higher rates of alcohol use disorder (AUD) compared to their parents.

## Social support, resilience, and quality of life

Scores for family and social support, presented in Table 5, did not show significant differences between WTC-wR and WTC-pR and were also comparable between parents and their children. Due to the lower rates of marriage or partnerships among adult children, there was substantial missing data for questions related to spouse/partner support in this

**Table 3. Exposure characteristics to the terror attack of the WTC-R parents separately for WTC-wR and WTC-pR groups.**

| | WTC-wR (N=93) | | WTC-pR (N=81) | | | |
|---|---|---|---|---|---|---|
| | N | % | N | % | p | missing |
| Arrival Time | | | | | 0.076 | 2 |
| 9/11-9/12 | 67 | 72.04 | 69 | 85.2 | | |
| 9/13-9/17 | 21 | 22.58 | 8 | 9.88 | | |
| 9/18 and later | 5 | 5.38 | 4 | 4.94 | | |
| Dust exposure | | | | | 0.099 | 5 |
| None | 44 | 47.83 | 25 | 31.65 | | |
| Some, not in cloud | 23 | 25.00 | 26 | 32.91 | | |
| In cloud | 25 | 27.17 | 28 | 35.44 | | |
| Ever worked on the pile/in the pit | 48 | 52.17 | 47 | 61.04 | 0.250 | 7 |
| Was exposed to human remains | 44 | 53.01 | 54 | 72.97 | **0.010** | 17 |
| Was South of Canal St. on 9/11 | 49 | 52.69 | 56 | 70.00 | **0.020** | 3 |
| | **Mean** | **SD** | **Mean** | **SD** | | |
| Level of exposure (range, 0–3) | 1.17 | 0.68 | 1.42 | 0.76 | **0.003** | 8 |
| Total exposure (months) | 3.93 | 3.83 | 4.61 | 3.16 | 0.206 | 7 |
| Hours at the site | | | | | | |
| On Days 1–2 | 14.58 | 12.68 | 20.46 | 12.46 | **0.002** | |
| On Days 3–7 | 43.15 | 28.29 | 46.89 | 28.00 | 0.383 | |
| On Days 8–20 | 96.23 | 75.09 | 107.92 | 73.64 | 0.302 | |

**Table 4. Mental health characteristics of the WTC-R parents and their children, stratified by WTC-wR and WTC-pR groups.**

| | Parent | | | | | | Adult Children | | | | | |
|---|---|---|---|---|---|---|---|---|---|---|---|---|
| | WTC-wR | | WTC-pR | | | | WTC-wR | | WTC-pR | | | |
| | Mean | SD | Mean | SD | p | missing | Mean | SD | Mean | SD | p | missing |
| Life Traumatic Events | 10.22 | 4.73 | 12.96 | 3.88 | **<.0001** | | 8.69 | 4.86 | 9.82 | 4.63 | **0.060** | 1 |
| | N | % | N | % | | | N | % | N | % | | |
| Depression | 31 | 33.33 | 23 | 28.40 | 0.480 | | 29 | 21.64 | 25 | 21.01 | 0.902 | |
| Anxiety disorder | 26 | 27.96 | 15 | 18.52 | 0.140 | | 30 | 22.39 | 34 | 28.57 | 0.259 | |
| PTSD (vs. negative) | | | | | 0.730 | 26* | | | | | 0.578 | 101* |
| Possible PTSD | 14 | 18.92 | 18 | 24.33 | | | 6 | 6.98 | 9 | 11.54 | | |
| PTSD | 13 | 17.57 | 12 | 16.22 | | | 6 | 6.98 | 6 | 7.69 | | |
| AUD | 8 | 8.60 | 7 | 8.64 | 0.990 | | 21 | 15.67 | 24 | 20.34 | 0.334 | 1 |
| SUD | | | 1 | 1.24 | 0.280 | | 3 | 2.24 | 1 | 0.84 | 0.373 | |
| Panic | | | | | | | | | | | | |
| score>8.75 | 28 | 30.11 | 25 | 30.86 | 0.910 | | 38 | 28.36 | 32 | 26.89 | 0.795 | |
| Not of psychiatric origin | 8 | 8.60 | 5 | 6.17 | 0.540 | | 5 | 3.73 | 9 | 7.56 | 0.183 | |

\* Missing values due to technical failure during data collection. Possible PTSD was defined as scores between 31 and 43, whereas PTSD was defined as scores ≥44.

group. Similarly, resilience and quality of life scores appeared consistent across responder groups and age groups. Quality of life was reported slightly lower among parents.

**Table 5. Social support, relationships, quality of life, and resilience scores of the WTC-R parents and their children stratified by WTC-wR and WTC-pR group.**

| | Parent | | | | | | Adult Children | | | | | |
|---|---|---|---|---|---|---|---|---|---|---|---|---|
| | WTC-wR | | WTC-pR | | | | WTC-wR | | WTC-pR | | | |
| | Mean | SD | Mean | SD | p | missing | Mean | SD | Mean | SD | p | missing |
| PARQ | | | | | | | | | | | | |
| Negative subscale | 7.39 | 2.69 | 7.24 | 2.32 | 0.702 | 2 | 9.45 | 3.46 | 9.00 | 3.49 | 0.303 | 7 |
| Positive subscale | 16.01 | 3.31 | 16.78 | 3.06 | 0.113 | 2 | 16.11 | 3.07 | 16.55 | 3.37 | 0.287 | 8 |
| General support score | 8.74 | 3.34 | 9.01 | 2.87 | 0.568 | 1 | 9.50 | 2.79 | 10.21 | 2.40 | **0.032** | 10 |
| Perceived Social Support | | | | | | | | | | | | |
| Spouse/partner support | 1.36 | 0.53 | 1.42 | 0.54 | 0.572 | 33# | 1.25 | 0.39 | 1.19 | 0.34 | 0.361 | 107# |
| Spouse/partner strain | 2.81 | 0.72 | 2.79 | 0.64 | 0.866 | 33# | 3.11 | 0.65 | 3.12 | 0.62 | 0.947 | 107# |
| Family support | 1.68 | 0.64 | 1.6 | 0.58 | 0.393 | 2 | 1.48 | 0.44 | 1.50 | 0.50 | 0.785 | 9 |
| Family strain | 2.95 | 0.71 | 2.89 | 0.65 | 0.273 | 2 | 2.71 | 0.66 | 2.77 | 0.71 | 0.487 | 9 |
| Friend support | 1.92 | 0.73 | 1.8 | 0.77 | 0.324 | 3 | 1.60 | 0.59 | 1.56 | 0.62 | 0.573 | 12 |
| Friend strain | 3.00 | 0.59 | 3.16 | 0.54 | 0.654 | 3 | 3.13 | 0.53 | 3.22 | 0.55 | 0.199 | 13 |
| Resilience | 8.45 | 1.43 | 8.35 | 1.66 | 0.670 | 1 | 8.24 | 1.45 | 8.14 | 1.45 | 0.586 | 10 |
| Quality of life | 47.32 | 10.8 | 48.35 | 8.96 | 0.495 | 1 | 52.13 | 9.11 | 51.31 | 9.64 | 0.494 | 9 |

Perceived Support scoring: Higher support scores indicate lower perceived support, and higher strain scores indicate lower perceived support.

#Indicating a marital status of being single.

## Factors associated with adult-children's mental health outcomes

We tested whether parental mental health was associated with that of their adult children and whether parental occupation category moderated these associations. As shown in Figs 2 and 3 (S1 Table), we analyzed the relationships of parental 9/11 exposure, occupation, and mental and physical health status with each of the following psychiatric outcomes in their adult children: depression, anxiety, panic disorder, PTSD, and alcohol use disorder. SUD was excluded due to its low prevalence in the sample. COVID-related PTSD symptoms (COVID-PCL scores), which were not significantly associated with any exposures, are not presented in the figures but are reported in the SM.

Among the parental exposure variables (Fig 2), earlier arrival at the WTC site, exposure to human remains, and greater cumulative hours spent at the site during the first 20 days post-9/11 were significantly associated with an increased risk of PTSD in their now-adult children. Similarly, extended time spent at the site in the early weeks following the attacks was linked to elevated risk for anxiety and panic among the now-adult children. In terms of current parental mental health (Fig 3), parental PTSD was significantly associated with higher odds of PTSD, panic, and depression in their adult children. Parental depression was associated with a higher likelihood of depression in their children, whereas parental panic disorder was associated with a higher likelihood of PTSD in their children. Additionally, a greater number of parental traumatic life events was associated with an increased likelihood of depression in the adult child (see S1 Table). Notably, the specific occupational role of the WTC responder was not significantly associated with any mental health outcomes in their children. To assess the generalizability of our findings, we conducted additional analyses using models weighted to account for the probability of participation relative to the originally invited WTC-R cohort. The results were highly consistent (although not identical) with our primary analyses, suggesting that the study sample is qualitatively representative of the broader WTC-R population. Full results from the weighted models are presented in the S2 Table.

Social support in general and from parents, family, or friends was associated with adult children's current mental health, as shown in Fig 4 (S3 Table). Specifically, higher negative parent-child relationship (PARQ) scores were linked to increased symptoms of depression and AUD in the adult-children. Lower positive PARQ scores were related to a

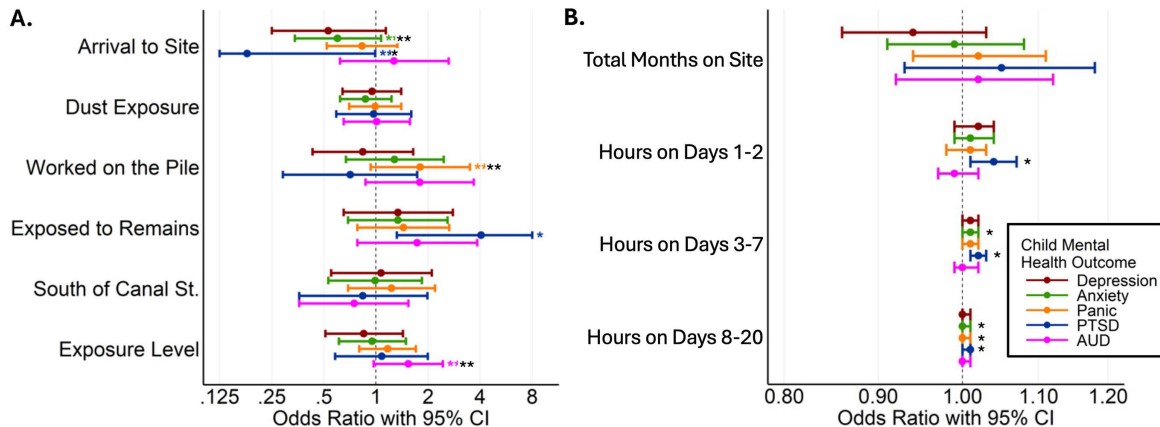

**Fig 2. Association between WTC-R parental exposure measures with now-adult children's current mental health adjusted for child sex, age, race/ethnicity, and parents' sex. A.** Type and location of exposure, and **B.** Exposure duration. Each parent's 9/11 exposure or mental health factor was tested separately. * p ≤ 0.05, and ** 0.05 < p < 0.1.

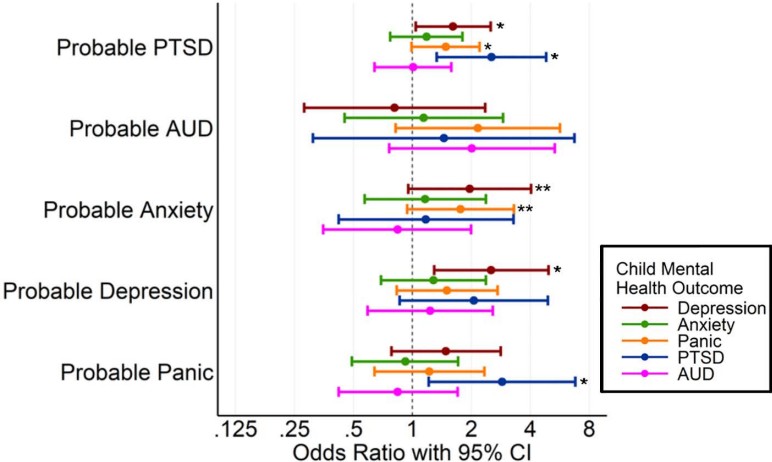

**Fig 3. Association between WTC-R parental mental health measures with now-adult children's current mental health adjusted for child sex, age, race/ethnicity, and parents' sex.** Each parent's 9/11 exposure or mental health factor was tested separately. * p ≤ 0.05, and ** 0.05 < p < 0.1.

greater likelihood of anxiety. Adult children were more likely to report PTSD when their parents reported a lack of general social support or strained relationships with a spouse. Furthermore, lower parental partner support was associated with an increased likelihood of depression in their children. Depression in adult children was also linked with strained family relationships and lower parental quality of life. Family strain was also associated with a higher likelihood of AUD in adult children. Notably, parental resilience was not associated with children's mental health problems.

The significant interactions between adult children's outcomes, parental exposure, social factors, and occupational groups are presented in Table 6 for both weighted and unweighted models. The unweighted model results indicate notable differences in how parental exposure affects adult children's mental health, depending on the parent's occupational group. Specifically, anxiety disorders were more likely among adult children of WTC-wR and less likely among adult children of WTC-pR when the parent had spent time at the pit or debris site. Adult children of WTC-pR also showed a lower likelihood

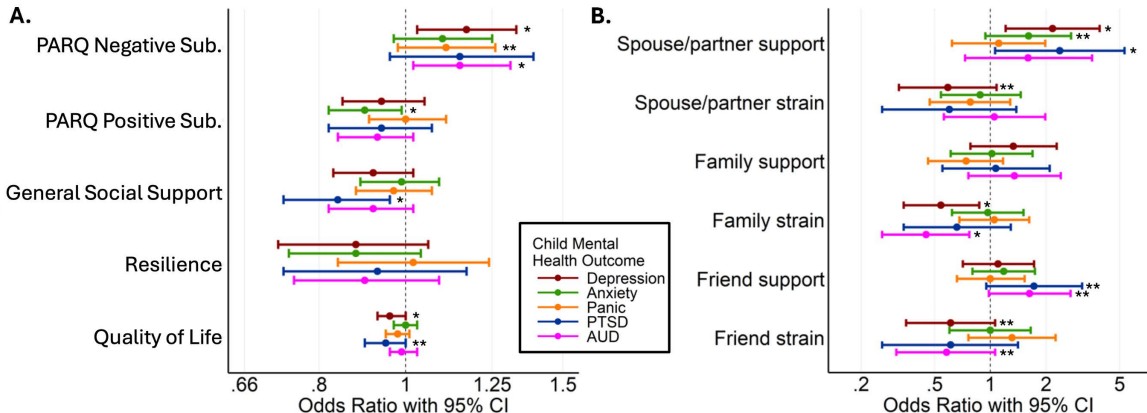

**Fig 4. Association between WTC-R parental social support measures and current mental health of the now-adult children, adjusted for child sex, age, race/ethnicity, and parents' sex and occupational category.** Each parent's social support factor was tested separately. **A**. Parent-child relationship, general support, and functioning. **B**. Family and friends' relationship with parents. * p ≤ 0.05, and ** 0.05 < p < 0.1.

**Table 6. Association of children's mental health with parent 9/11 exposure, current mental health, and social support by parent occupational status, adjusted for child sex, age, minoritized race/ethnicity, and parent sex.**

| Offspring outcome | Parental exposure | Interaction p | WTC-wR | | | WTC-pR | | |
|---|---|---|---|---|---|---|---|---|
| | | | OR | 95% CI | p | OR | 95% CI | p |
| *Unweighted models* | | | | | | | | |
| Anxiety | Ever worked on the pile/in the pit | 0.001 | 3.98 | [1.39,11.37] | **0.010** | 0.40 | [0.16,0.98] | **0.044** |
| PTSD | Ever worked on the pile/in the pit | 0.003 | 2.71 | [0.72,10.23] | 0.142 | 0.18 | [0.05,0.66] | **0.010** |
| PTSD | Life Traumatic events | 0.036 | 1.21 | [1.04,1.42] | **0.014** | 0.98 | [0.87,1.11] | 0.778 |
| Depression | Partner strain | 0.005 | 0.21 | [0.08,0.56] | **0.002** | 1.26 | [0.58,2.76] | 0.561 |
| AUD | PARQ Negative | 0.038 | 0.99 | [0.80,1.23] | 0.924 | 1.31 | [1.11,1.54] | **0.001** |
| Panic | Resilience | 0.028 | 1.30 | [0.96,1.76] | 0.091 | 0.81 | [0.62,1.06] | 0.131 |
| *Weighted models* | | | | | | | | |
| Anxiety | Ever worked on the pile/in the pit | 0.002 | 5.13 | [1.66,15.90] | **0.005** | 0.50 | [0.19,1.31] | 0.157 |
| PTSD | Ever worked on the pile/in the pit | 0.001 | 4.26 | [1.12,16.18] | **0.033** | 0.19 | [0.05,0.76] | **0.018** |
| PTSD | Was exposed to human remains | 0.047 | 16.65 | [1.90,145.57] | **0.011** | 1.13 | [0.23,5.59] | 0.883 |
| Depression | Possible PTSD vs. none | 0.788 | 1.04 | [0.27,3.99] | 0.953 | 1.34 | [0.37,4.90] | 0.657 |
| | PTSD vs. none | 0.041 | 1.26 | [0.34,4.66] | 0.726 | 9.54 | [2.29,39.74] | **0.002** |
| Depression | Depression | 0.026 | 1.25 | [0.46,3.37] | 0.661 | 6.29 | [2.12,18.67] | **0.001** |
| PTSD | Life Traumatic events | 0.026 | 1.21 | [1.02,1.42] | **0.026** | 0.96 | [0.85,1.07] | 0.437 |
| Depression | Partner strain | 0.004 | 0.17 | [0.06,0.48] | **0.001** | 1.40 | [0.53,3.71] | 0.502 |
| PTSD | Partner strain | 0.035 | 0.20 | [0.05,0.80] | **0.023** | 1.68 | [0.43,6.59] | 0.460 |
| AUD | PARQ Negative | 0.020 | 1.00 | [0.83,1.20] | 0.997 | 1.36 | [1.14,1.62] | **0.001** |

of PTSD when their parents worked at the pit. In contrast, PTSD was more likely among adult children of WTC-wR when parents reported exposure to major life traumatic events and when parental depression co-occurred with strained spousal or partner relationships. Among adult children of WTC-pR, the likelihood of AUD increased when they reported negative relationships with their parents, as measured by the PARQ scores. While a significant interaction was observed between parental occupational group and parental resilience in relation to child panic, neither stratum-specific estimate reached statistical significance. Weighted models produced patterns generally consistent with unweighted models. In addition,

weighted estimates indicated that adult children of WTC-wR were more likely to have PTSD when their parents worked at the pit, were exposed to human remains, or reported more life-traumatic events. In contrast, adult children of WTC-pR were more likely to have depression when their parents' reports were indicative of PTSD or depression.

## Discussion

The present study provides new evidence for ITT among families affected by the WTC terror attacks. More than two decades after 9/11, we found that both parental exposure history and recent parental mental health were significantly associated with their now-adult children's current psychological outcomes. Specifically, greater parental exposure, indexed by earlier arrival at the WTC site, longer time spent at the debris pile, and contact with human remains, was associated with higher risk for PTSD, anxiety, and panic symptoms among adult offspring. Similarly, current parental PTSD, anxiety, or depression was associated with adult children's PTSD, depression, and panic. These findings support the notion that the enduring psychological consequences of extremely adverse events extend beyond directly exposed individuals and are consistent with prior evidence of intergenerational transmission of large-scale traumatic events [3]. However, our study differs from previous investigations of intergenerational effects of the 9/11 disaster in several important ways. First, our sample focused exclusively on parents who were WTC responders, whereas prior studies included both responders and civilian survivors [23,45]. Second, the majority of the adult children in our sample were not directly exposed to the 9/11 attacks, in contrast to earlier studies in which many participating children had been personally exposed to the event or its aftermath [43,46]. This distinction allows us to better demonstrate the indirect effects of responders' exposure and mental health on offspring outcomes, thereby providing stronger evidence for intergenerational transmission of trauma independent of shared exposure.

Importantly, our study extends prior research [19] documenting the influence of parental mental health on children's emotional and behavioral regulation in the non-WTC-R population by demonstrating that these associations persist into adulthood. This finding underscores the enduring developmental consequences of parental trauma exposure and mental health burden among offspring of WTC responder families. As many of these adult children are now forming their own families, the potential for these intergenerational effects to extend to a third generation warrants attention [47,48].

Our results also point to the role of family and social relationships as potential mediators of these intergenerational effects. Poor parent–child relationships, characterized by higher negativity and lower perceived warmth, were associated with increased symptoms of depression, anxiety, and AUD among adult children. Similarly, offspring whose parents reported strained spousal relationships were more likely to exhibit depression and PTSD. These findings align with theoretical models emphasizing the role of relational and social environmental pathways in transmitting trauma across generations [49,50]. Social support and family cohesion are well-documented protective factors that can buffer the psychological effects of trauma [51]. The current study reinforces that relational dynamics within the family may either mitigate or amplify the impact of parental trauma exposure on offspring's well-being. In contrast, a strained friendship or a lack of friendship support among parents did not appear to significantly negatively impact their children's mental health outcomes.

Our analyses further revealed that occupational role moderated several intergenerational associations between parental trauma and offspring mental health. Among WTC-wR, greater exposure severity and cumulative life stressors were associated with elevated anxiety and PTSD symptoms in their adult children. In contrast, among WTC-pR, negative parent–child relationships were linked to higher rates of alcohol use disorder (AUD) in offspring. These occupational differences likely reflect distinct professional cultures, stress exposures, and coping mechanisms between police and non–law enforcement responders.

First responder occupations, particularly within law enforcement, are characterized by strong cultural norms that emphasize emotional control, stoicism, and self-reliance [52]. Such occupational values may buffer against the overt expression of distress, potentially protecting against the development of PTSD and depression in responders themselves [53]. However, these same coping patterns, particularly emotional suppression and avoidance, have been linked to higher

rates of maladaptive behaviors, including alcohol misuse and interpersonal difficulties [54]. Indeed, epidemiological studies consistently report that police officers exhibit a higher prevalence of alcohol use disorders compared to other first responders and the general population [55]. From an ITT perspective, such occupationally reinforced coping norms may influence parenting styles and emotional communication within the family. Children raised in these environments may internalize similar patterns of emotional inhibition or reliance on external coping strategies, thereby increasing their vulnerability to certain psychopathological outcomes such as anxiety or SUD. This pattern was observed in our data, where only the offspring of WTC-pR demonstrated an increased likelihood of AUD when parent–child relationship quality was poor. Thus, while first responder culture may provide psychological resilience and a sense of purpose that mitigates direct trauma-related psychopathology, it may also shape family dynamics in ways that differentially influence offspring risk for mental health disorders. These findings underscore the importance of accounting for occupational context and associated sociocultural norms in models of ITT.

Despite being selected for PTSD following 9/11, WTC-R parents in our sample reported relatively low current PTSD prevalence, alongside elevated rates of depression and anxiety. This finding is consistent with longitudinal studies of 9/11 samples showing that while acute PTSD symptoms tend to diminish over time [56], many individuals experience a chronic or delayed emergence of other internalizing disorders, such as depression and generalized anxiety [57]. For example, longitudinal analyses of the World Trade Center Health Registry have documented a gradual decline in PTSD prevalence from approximately 19 – 20% in the first decade post - 9/11 to about 13–14% 15 years later, while comorbid depression and anxiety symptoms have remained stable or increased [57,58]. This symptom evolution may reflect both natural recovery processes and the cumulative psychological toll of aging, occupational stress, and other life events, which can reshape the manifestation of trauma-related distress over time [10,59]. Our data suggest that interventions aimed at reducing PTSD symptoms should also target comorbid mental health conditions, such as depression and anxiety. In the context of ITT, prolonged recovery trajectories and untreated comorbidities may heighten the risk of adverse outcomes in offspring. Therefore, early and comprehensive intervention to alleviate posttraumatic symptoms in parents may help prevent or reduce the transmission of trauma-related distress to the next generation.

Taken together, these findings contribute to a growing body of literature supporting multi-level models of ITT. Such models emphasize that biological, psychological, and social mechanisms interact to shape vulnerability and resilience across generations [60,61]. The associations between parental trauma exposure, psychopathology, and offspring outcomes observed here may reflect shared environmental stressors, learned emotional regulation patterns, or epigenetic modifications [62]. Our data suggest that interventions targeting parental well-being and family relationships could reduce the propagation of trauma-related distress to subsequent generations.

Several limitations warrant consideration. First, all mental health assessments relied on self-reported screening instruments rather than structured clinical diagnostic interviews, which may have led to either over- or underestimation of the true prevalence of psychiatric conditions. Second, participation bias is possible, as families who agreed to take part in the study may differ systematically from those who declined. For example, participating families may have stronger family cohesion, greater awareness of mental health issues, or higher motivation to engage in research, whereas families who declined participation may experience greater family dysfunction or more severe mental health conditions that hinder their ability to participate. Although we applied statistical weighting procedures to improve the representativeness of the sample, these adjustments were primarily based on demographic variables and did not account for all mental health characteristics (only PTSD) or exposure-related variables. Third, the sample size, while adequate for detecting moderate effects, limits statistical power for subgroup analyses and the detection of smaller associations. Future longitudinal studies, ideally incorporating multimethod assessments and clinical interviews, are needed to clarify mechanisms of ITT and to inform interventions aimed at protecting future generations from the enduring impact of trauma.

Despite these limitations, this study offers compelling evidence that the psychological impact of the WTC disaster continues to reverberate across generations. By integrating parental exposure characteristics, mental health, and family

relationships, it highlights the complex and enduring pathways through which trauma legacies shape offspring's mental health. These findings highlight the need for trauma-informed, family-centered interventions that address both parental and child well-being, as well as continued surveillance of WTC-exposed families as they age.

## Supporting information

**S1 Table. Results of the unweighted model represented in Figs 2 and 3.**
(PDF)

**S2 Table. Results of weighted model.**
(PDF)

**S3 Table. Results of the unweighted model represented in Fig 4.**
(PDF)

**S1 Text. Social Support (3 items from MOS).**
(PDF)

**S2 Text. COVID questionnaire.**
(PDF)

**S1 Appendix. Tables Tables A through E show overall sample information, expanding Tables 2–5.**
(PDF)

## Acknowledgments

We are deeply grateful to all study participants for their time, trust, and willingness to share their experiences. We extend our sincere appreciation to Dr. Musa for his invaluable support in establishing the REDCap database and for his ongoing technical assistance throughout data collection. We want to express our gratitude to Drs. Adriana Feder and Robert J. Pietrzak for their contributions to the conceptualization and early development of this study. We also thank the GRDC for providing access to essential data resources. Finally, we extend our gratitude to the research coordinator, Sydney Solomon, and our dedicated student interns, whose outreach efforts and commitment to engaging World Trade Center responders were essential to recruitment and study completion. This work would not have been possible without each of you.

## Author contributions

**Conceptualization:** Yael M. Cycowicz, Diana V. Rodriguez-Moreno, Keely Cheslack-Postava.

**Data curation:** Yael M. Cycowicz.

**Formal analysis:** Yael M. Cycowicz, Keely Cheslack-Postava.

**Funding acquisition:** Yael M. Cycowicz, Diana V. Rodriguez-Moreno, Keely Cheslack-Postava.

**Investigation:** Diana V. Rodriguez-Moreno, Daniel Craft, Keely Cheslack-Postava.

**Methodology:** Yael M. Cycowicz, Diana V. Rodriguez-Moreno, Daniel Craft, Keely Cheslack-Postava.

**Project administration:** Yael M. Cycowicz.

**Supervision:** Yael M. Cycowicz, Daniel Craft.

**Validation:** Yael M. Cycowicz.

**Visualization:** Diana V. Rodriguez-Moreno.

**Writing – original draft:** Yael M. Cycowicz.

**Writing – review & editing:** Yael M. Cycowicz, Diana V. Rodriguez-Moreno, Keely Cheslack-Postava.

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
