## [Decision Letter · Decision Letter 0]

24 Feb 2026

The Long Shadow of 9/11: Mental Health Outcomes in Adult Children of World Trade Center Responders with PTSD

PMEN-D-26-00044

Dear Prof. Cycowicz,

We are pleased to inform you that your manuscript 'The Long Shadow of 9/11: Mental Health Outcomes in Adult Children of World Trade Center Responders with PTSD' has been provisionally accepted for publication in PLOS Mental Health.

Best regards,

Vitalii Klymchuk, Ph.D., D.Sc.

Academic Editor

PLOS Mental Health

Reviewer Comments (if any, and for reference):

Reviewer's Responses to Questions

**Comments to the Author**

1. Does this manuscript meet PLOS Mental Health’s publication criteria? Is the manuscript technically sound, and do the data support the conclusions? The manuscript must describe methodologically and ethically rigorous research with conclusions that are appropriately drawn based on the data presented.? Is the manuscript technically sound, and do the data support the conclusions? The manuscript must describe methodologically and ethically rigorous research with conclusions that are appropriately drawn based on the data presented.

Reviewer #1: Yes

Reviewer #2: Yes

2. Has the statistical analysis been performed appropriately and rigorously?

Reviewer #1: Yes

Reviewer #2: Yes

3. Have the authors made all data underlying the findings in their manuscript fully available (please refer to the Data Availability Statement at the start of the manuscript PDF file)?

The PLOS Data policy requires authors to make all data underlying the findings described in their manuscript fully available without restriction, with rare exception. The data should be provided as part of the manuscript or its supporting information, or deposited to a public repository. For example, in addition to summary statistics, the data points behind means, medians and variance measures should be available. If there are restrictions on publicly sharing data—e.g. participant privacy or use of data from a third party—those must be specified.requires authors to make all data underlying the findings described in their manuscript fully available without restriction, with rare exception. The data should be provided as part of the manuscript or its supporting information, or deposited to a public repository. For example, in addition to summary statistics, the data points behind means, medians and variance measures should be available. If there are restrictions on publicly sharing data—e.g. participant privacy or use of data from a third party—those must be specified.

Reviewer #1: Yes

Reviewer #2: No

4. Is the manuscript presented in an intelligible fashion and written in standard English?

Reviewer #1: Yes

Reviewer #2: Yes

5. Review Comments to the Author

Reviewer #1: After careful reading, I find this article very interesting. It meets the methodological criteria. The authors present the results in a clear and understandable way. I do not think any modifications are necessary. Congratulations

Reviewer #2: The paper is informative and well written. Some minor changes are required.

1. Add the figures in the results section so the reader can get the pictorial view along with the discussion part.

2. Ethical concerns regarding the hadeling of symptoms, if the study triggers previous disturbing thoughts should be addressed properly.

6. PLOS authors have the option to publish the peer review history of their article (what does this mean?). If published, this will include your full peer review and any attached files.). If published, this will include your full peer review and any attached files.

**Do you want your identity to be public for this peer review?** For information about this choice, including consent withdrawal, please see our Privacy Policy..

Reviewer #1: **Yes:**Vicente Robles AlonsoVicente Robles Alonso

Reviewer #2: No
